# Public awareness of colorectal cancer symptoms and risk factors, and exploring screening barriers across nine countries: A multi-national cross-sectional study

Aseel Ghanayem[1,2☯], Mandy Elewa[2,3], Husam Abu Dawood[2,4*],
Tarek A. Owais[2,5], Rawan Ghanayem[6], Hanan Khaled Almokdad[7], Nael Kamel Eltewacy[2,8☯],
EARG Collaborators[¶]

1 Faculty of Medicine, The University of Jordan, Amman, Jordan, 2 Eltewacy Arab Research Group (EARG), 3 Faculty of Pharmacy, Istanbul University, Istanbul, Türkiye, 4 Faculty of Medicine, Palestine Polytechnic University, Hebron, Palestine, 5 Faculty of pharmacy, Beni-suef University, Beni-suef, Egypt, 6 Faculty of Pharmacy, The University of Jordan, Amman, Jordan, 7 Faculty of Physical Therapy, Al Rasheed International Private University for Science and Technology, Damascus, Syria, 8 École Normale Supérieure Paris-Saclay, Paris, France

¶ Membership of EARG Collaborators is provided in Supporting Information file S1 Text
☯ These authors contributed equally to this work.
* Husamnoah11@gmail.com

## Abstract

Colorectal cancer is the second leading cause of cancer-related deaths worldwide and the third most common type of cancer overall. Colorectal cancer can be avoided with regular screening and lifestyle modifications. This study aimed to identify the barriers to screening for colorectal cancer and to evaluate the public's awareness and knowledge of the disease's symptoms and risk factors. A cross-sectional study was conducted across nine countries from February 1, 2024, through April 3, 2024, targeting individuals aged 18 years and older. Data were collected using an online self-administered questionnaire in collaboration with national leaders and local team members in each country. Logistic regression analyses were employed to examine factors associated with participants' knowledge, awareness, and screening barriers. Among the 13,030 participants, awareness of CRC symptoms was highest for "lump in the abdomen" (70%) and lowest for "back pain" (39%). The most recognized risk factor was "history of bowel disease" (71%), whereas the least recognized was "low fruit/vegetable intake" (33%). The primary barrier to screening was the belief in low personal risk due to the absence of symptoms (63%). Multivariate logistic regression identified working in the healthcare sector as the strongest predictor of high knowledge (aOR = 2.70, 95% CI: 2.50–2.94) relative to non-healthcare participants. Higher knowledge was also significantly associated with male sex, older age (25–49 years), and university-level education (all p < 0.001). Colorectal cancer usually goes unnoticed until it reaches an advanced stage, when treating it becomes far more difficult. These findings underscore an urgent need for targeted, culturally sensitive public

**Data availability statement:** All relevant data are within the paper and its Supporting Information files. The full anonymized dataset has also been deposited in Open Science Framework (OSF) at: https://doi.org/10.17605/OSF.IO/5FEHM.

**Funding:** The authors received no specific funding for this work.

**Competing interests:** The authors have declared that no competing interests exist.

health campaigns across the region to address knowledge gaps and mitigate barriers, particularly the prevalent misconception of asymptomatic low risk. These include avoiding risk factors, maintaining a healthy lifestyle, and participating in screening programs to detect disease early.

---

## 1. Introduction

Around the world, colorectal cancer (CRC) is the third most common type of cancer and the second leading cause of cancer-related deaths according to the World Health Organization (WHO, 2024) [1]. CRC primarily affects individuals aged 50 and above. In 2020, more than 1.9 million people were diagnosed with colorectal cancer worldwide, and over 930,000 lost their lives due to the disease. The number of new cases and annual fatalities from CRC is expected to rise by 63% by 2040, accounting for about 3.2 million cases [1]. Colorectal cancer (CRC) incidence is rising in low- and middle-income countries (LMICs), driven by demographic and economic transitions as well as changes in lifestyle and dietary habits, such as physical inactivity, tobacco and alcohol use, high consumption of red or processed meat, and low intake of fruits and vegetables, are associated with increased colorectal cancer risk. [2]. While many high-income countries have achieved significant declines in CRC incidence and mortality through structured national screening programs, LMICs continue to face a growing burden [2]. In these settings, screening efforts are often limited, sporadic, and largely reliant on individual initiative rather than organized programs [3]. A recent scoping review further highlights that most CRC screening activities in middle-income countries are relatively new, small in scale, and have not yet reached wide population coverage [4].

In more than 80% of CRC cases, a diagnosis is made after symptoms appear [5]. Both specific and non-specific signs, such as changes in bowel habits, blood in the stool, unexplained weight loss, and anemia, can serve as warning indicators of CRC [6]. Unfortunately, many individuals normalize these symptoms and delay seeking medical help, even in cases that have clear indications of colorectal cancer [7]. This emphasizes the importance of raising public knowledge about CRC symptoms in order to encourage early detection and prompt medical care.

Male gender, obesity, advanced age (50 years and older), a family history of hereditary colorectal cancer, specific racial origins, inflammatory bowel disease, diabetes, consuming large amounts of red or processed meat, smoking, and alcohol consumption are the main risk factors associated with colorectal cancer (CRC) [8]. Increasing awareness of these risk factors and symptoms can lead to earlier recognition and greater participation in screening programs [9,10]. Early diagnosis through screening programs can significantly reduce mortality and improve survival outcomes [11].

Although colorectal cancer (CRC) can be cured if detected early, unfortunately, many countries still lack well-established national screening programs [2]. American College of Gastroenterology (ACG) guidelines recommend colorectal cancer screening in average-risk individuals between ages 50 and 75 years, with a conditional suggestion to begin screening between ages 45 and 49 years; colonoscopy and fecal

immunochemical testing (FIT) are recommended as the primary screening modalities, while other tests (e.g., flexible sigmoidoscopy, multitarget stool DNA, CT colonography) may be considered when these are not possible.[12]. The USPSTF recently revised its colorectal cancer screening guidelines to include adults aged 45–49 years in the recommended screening population [13].

Even in countries with national screening programs and despite the clear benefits of colorectal cancer (CRC) screening, participation rates remain significantly lower compared to other recommended adult preventive programs [14]. Several barriers hinder individuals from undergoing regular colorectal cancer (CRC) screening. These include lack of awareness about CRC and its screening, absence of physician recommendation, fear or anxiety regarding the procedure or its results, embarrassment or cultural stigma, limited access to healthcare services, financial constraints, and misconceptions about the need for screening in the absence of symptoms [15]. This necessitates the identification of the profound issues and creating workable plans to encourage more participation in screening programs.

Several studies have examined colorectal cancer awareness and screening perceptions in individual countries, including work from UAE, Palestine, Malaysia, Jordan, Iran, and other countries in the region [16–20]. These studies consistently report limited knowledge of CRC symptoms and risk factors, along with multiple screening barriers such as fear, embarrassment, lack of physician recommendation, and low perceived risk. However, most available evidence is country-specific and focuses on single populations, which limits the ability to compare awareness and barriers across different cultural and healthcare contexts.

To our knowledge, no previous study has conducted a large multi-country assessment of public awareness, knowledge, and screening barriers for colorectal cancer across several low- and middle-income countries using a unified tool. Addressing this gap is essential to better understand similarities and differences across populations and to support regionally informed public health strategies.

This study aims to comprehensively evaluate public awareness of colorectal cancer symptoms and risk factors, and exploring screening barriers across nine countries (Jordan, Egypt, Iraq, Bahrain, Syria, Pakistan, Palestine, Yemen, and Nigeria). Additionally, it seeks to identify the barriers that prevent individuals from undergoing CRC screening. Understanding these factors will help healthcare institutions enhance public awareness of colorectal cancer and develop tailored screening programs that effectively address these barriers, thereby increasing participation in screening initiatives.

## 2. Methods

### 2.1. Design

A multi-national, cross-sectional study among the general population in nine countries (Jordan, Egypt, Iraq, Bahrain, Syria, Pakistan, Palestine, Yemen, and Nigeria) between February 1st, 2024, and April 3rd, 2024 was conducted, using an online validated questionnaire.

### 2.2. Eligibility criteria

Individuals aged 18 years or older were eligible to participate, as adults are legally able to provide informed consent and colorectal cancer awareness, attitudes, and screening behaviors are most relevant in this population. Participants were required to be able to complete the questionnaire in English or Arabic. Exclusion criteria included a personal history of colorectal cancer, inability to provide informed consent, previous completion of the survey, or submission of incomplete responses.

### 2.3. Sampling

A convenience sampling approach was used. The minimum required sample size of 385 participants per country was calculated using the Raosoft [21]. A total of 16,940 individuals were invited to participate across the nine study countries. Of these, 3,910 responses were excluded due to ineligibility (age < 18 years, residency outside the targeted countries, or duplicate entries) or incomplete surveys, resulting in a final analytical sample of 13,030 participants.

Countries that achieved ≥385 respondents were included in country-specific analyses, while countries with fewer than 385 respondents were not reported individually; however, their valid responses were retained in the pooled multi-national analysis.

The study initially targeted nine countries (Jordan, Egypt, Iraq, Syria, Yemen, Bahrain, Pakistan, Palestine, and Nigeria). These countries were selected based on geographical representation across the Middle East, North Africa, and Asia; feasibility of recruitment; and the presence of local collaborators to facilitate participant outreach. During data collection, additional eligible responses were received from Bangladesh, Saudi Arabia, Sudan, and the United Arab Emirates. These responses were retained in the pooled multinational analysis; however, due to small sample sizes, they were not reported individually in country-specific comparisons.

### 2.4. Questionnaire

The awareness component was derived from the Colorectal Cancer Awareness Measure (Colorectal CAM), as used and validated in the study by Al-Dahshan et al. [22]. The barriers component was obtained from a subsequent study by the same authors that developed and translated the barriers items to assess perceived screening barriers in Qatar [23]. The Arabic versions of both components were used without modification, and their prior use in comparable populations supports their validity for our study.

We created a self-administered survey using Google Forms, offering it in both English and Arabic so participants could choose the language they felt most comfortable with.

The questionnaire was divided into four sections:

1. Socio-demographic data: Age, gender, country of residency, educational level, marital status, working in the healthcare sector, and current occupational status.

2. Knowledge and awareness of Colorectal Cancer symptoms: consisted of 9 questions about the symptoms of colorectal cancer. Respondents were presented with nine symptoms and asked to indicate whether they believed each was an indicator of colorectal cancer, with response options including "Yes," "No," or "Don't know."

3. Knowledge and awareness of colorectal cancer risk factors: 10 inquiries were made concerning risk factors associated with colorectal cancer. Participants were presented with ten different risk factors and asked whether they considered each as an indicator of colorectal cancer, with response options including "Yes," "No," or "Don't know."

4. Colorectal cancer screening barriers: the participants were asked to what extent they think the following eleven factors can form barriers to undertaking annual colorectal screening. Response options included "strongly agree", "somewhat agree ", "neutral ", "somewhat disagree ", "strongly disagree ", or "don't know".

### 2.5. Data collection

To ensure high-quality data collection, we designated national leaders in each country to oversee the process and obtain ethical approvals. These leaders, in turn, recruited two to five collaborators to assist in gathering the required sample. Participants were recruited using a convenience sampling approach. In each participating country, national study leaders and local collaborators distributed the online self-administered questionnaire through social media platforms, email networks, and direct communication channels. Participation was voluntary, and all adults aged ≥18 years were invited to participate. Because recruitment was conducted online, individuals without internet access, limited digital literacy, or those who were illiterate were less likely to participate, which may have introduced selection bias.

An online Google Form was distributed for data collection. The online survey link collected data anonymously, without recording any personal or contact information. Before starting the questionnaire, participants were asked to either provide

their consent or decline participation. Those who consented were asked to select their preferred language—Arabic or English. Subsequently, two confirmatory questions were included: the first to confirm eligibility for participation, and the second asking whether the respondent had previously completed the survey for this study, participants indicating prior participation were automatically excluded, thereby preventing duplicate entries. We excluded incomplete responses to reduce the risk of information bias. The online questionnaire data were automatically collected in an Excel spreadsheet. Responses completed in Arabic were translated into English and then combined with the English responses for analysis.

A total of 16,940 individuals accessed the questionnaire. Of these, 3,910 entries were excluded due to ineligibility (e.g., age below 18 years), incomplete responses, or duplicate submissions, resulting in a final analytical sample of 13,030 participants.

## 2.6. Ethical considerations

Ethical approval was obtained from the relevant ethics committees in Egypt (Al Azhar Faculty of Medicine), Yemen (Al-Razi University; Ref: 01/FMHS/2024), Iraq (Ministry of Health, Busra; Ref: 543), and Syria (Al Rasheed International Private University for Science and Technology). In Jordan, Bahrain, Pakistan, Palestine, and Nigeria, data were collected using an anonymous, self-administered online survey and were considered minimal-risk research under local regulations; therefore, additional local ethics approval was not required in these countries according to national guidance for anonymous survey-based studies. All study procedures were conducted in accordance with the Declaration of Helsinki. In addition, written consent was obtained from the participants after a detailed explanation of the study's purpose before filling out the questionnaire, emphasizing their confidentiality and the complete preservation of their data.

## 2.7. Data analysis

Statistical analysis was performed using R statistical software version 4.1.3. Categorical variables, including demographic characteristics and occupation, were summarized using frequencies and percentages. Depending on the objective of each analysis, categorical variables were treated as either independent or dependent variables.

Knowledge items were scored dichotomously (1 = correct, 0 = incorrect), and a total knowledge score was calculated for each participant. As the Colorectal Cancer Awareness Measure (CAM) does not prescribe a specific cut-off for categorizing awareness levels, a ≥ 70% score was selected a priori to define "high knowledge." This threshold was chosen as a pragmatic benchmark reflecting attainment of most knowledge items and is consistent with conventions used in public health knowledge assessments. To assess the robustness of this classification, sensitivity analyses were conducted using alternative thresholds (≥60% and ≥80%), and the direction and significance of the associations remained unchanged. The knowledge categories were summarized using frequencies and percentages, while the chi-square test was applied to examine the relationship between each knowledge question and occupation.

Both univariate and multivariate logistic regression models were used to examine factors associated with overall knowledge level (low vs. high), with results presented as Odds Ratios (OR) and Adjusted Odds Ratios (aOR) with 95% confidence intervals (CI). Model adequacy was evaluated prior to interpretation; goodness-of-fit was assessed using the Hosmer–Lemeshow test and Akaike Information Criterion (AIC), and multicollinearity among predictors was examined using variance inflation factors (VIF). No evidence of poor fit or problematic multicollinearity was detected, and all model assumptions were met. Participants were grouped into three age categories: < 25, 25–49, and ≥50 years. This grouping was chosen to capture broad demographic differences in colorectal cancer knowledge across younger adults, middle-aged adults, and those approaching or within the previously recommended screening-eligible population (50–75 years) allowing for meaningful assessment of knowledge trends relevant to public health interventions.

Regarding barriers to screening, responses were reported using frequencies and percentages. The chi-square test with a 95% CI was used to analyze the association between reported barrier questions and occupation. A p-value of ≤ 0.05 was considered significant.

## 3. Results

### 3.1. Response rate and demographics

A total of 16,940 participants were invited to participate in the study. However, 3,910 responses were excluded due to ineligible demographics (age restrictions, residency outside targeted countries, or duplicate entries), resulting in a final sample size of 13,030 participants.

The study initially targeted nine countries. Additional responses from Bangladesh, Saudi Arabia, Sudan, and the United Arab Emirates were received and included in the overall pooled dataset; however, these countries were not analyzed individually due to smaller sample sizes.

Analysis of demographics revealed variations in response rates across countries. Syria had the highest response rate (14%), followed by Yemen and Pakistan (both 13%). Regarding occupation, approximately 37% (4,855 participants) work in the healthcare sector. The median age was 24 (IQR: 21–32). Males comprised 43% (n = 5,567) of the participants, while females accounted for 57% (n = 7,463), and 84% (n = 10,958) had university degrees. The marital status distribution included 32% married and 63% single. The employment status breakdown showed 53% unemployed and 2.6% retired. Age was significantly associated with occupation (p = 0.027), while all other demographic variables exhibited highly significant associations with occupation (p < 0.001), as shown in Table 1.

### 3.2. Knowledge regarding colon cancer

A variation in the level of awareness of CRC symptoms was reported as shown in (Table 2) Lump in the abdomen was the most recognized symptom (70%, n = 9,134), followed by unexplained weight loss (67%, n = 8,754) and bleeding from the back passage (65%, n = 8,526), whereas back pain had the lowest awareness level (39%, n = 5,083). (Fig 1). Regarding risk factors (RFs), the knowledge level also showed variations. S1 Table shows that the least recognized RF was consuming less than five portions of fruits and vegetables daily (33%, n = 4,246). In contrast, only 51% of participants (n = 6,709) were aware of the risks associated with red or processed meat consumption. Similar awareness levels were observed for a low-fiber diet (51%, n = 6,709). The highest awareness (71%, n = 9,270) was associated with having a history of bowel disease (e.g., ulcerative colitis or Crohn's disease). Alcohol consumption was also associated with a high awareness rate (70%, n = 9,152). Refer to (Fig 2) for a summarized list.

Notably, only 31% (n = 3,982) of participants demonstrated a high level of knowledge regarding both colon cancer symptoms and risk factors. Although country-specific breakdowns were not performed, the included countries share broadly similar cultural practices, dietary habits, and healthcare systems, which supports the use of a pooled analysis.

Knowledge levels regarding colorectal cancer differed significantly between participants working or studying in healthcare and those not in healthcare (p < 0.001; Cramér's V ≈ 0.25). Among healthcare participants, 46% demonstrated high knowledge compared with only 21% of non-healthcare participants, while 54% of healthcare participants and 79% of non-healthcare participants exhibited low knowledge. These results indicate a moderate association between occupation in the healthcare field and knowledge levels.

### 3.3. Barriers associated with colon screening

The barriers to colorectal cancer screening are presented in Table 3 and (Fig 3). The most commonly reported barriers were the belief that a healthy lifestyle reduces the need for screening (65%, n = 8,396) and the belief that screening is unnecessary when no symptoms are present (63%, n = 8,164). Psychological barriers were also common, including fear of receiving a cancer diagnosis (61%, n = 7,873), fear of the screening test (60%, n = 7,788), and embarrassment (58%, n = 7,464). In addition, 55% (n = 7,174) reported that having no family history made them feel screening was not necessary. Practical barriers were also noted, including perceived inconvenience (54%, n = 7,094), lack of reminders (52%, n = 6,715), and lack of time (47%, n = 6,195). All barriers were significantly associated with working or studying in the healthcare field

**Table 1. Demographic Characteristics of the Sample Population.**

| Characteristic | N | Overall (N = 13,030) (%) | Working in the health care sector (N = 4,855) (%) | Not working in the health care sector (N = 8,175) (%) | p-value |
|---|---|---|---|---|---|
| Age | 13,030 | 24 (21, 32) | 24 (22, 29) | 24 (20, 34) | 0.027* |
| Gender | 13,030 | | | | <0.001** |
| Male | | 5,567 (43%) | 2,236 (46%) | 3,331 (41%) | |
| Female | | 7,463 (57%) | 2,619 (54%) | 4,844 (59%) | |
| Country of residence | 13,030 | | | | <0.001** |
| Bahrain | | 912 (7.0%) | 347 (7.1%) | 565 (6.9%) | |
| Bangladesh | | 256 (2.0%) | 134 (2.8%) | 122 (1.5%) | |
| Egypt | | 1,513 (12%) | 577 (12%) | 936 (11%) | |
| Iraq | | 1,438 (11%) | 646 (13%) | 792 (9.7%) | |
| Jordan | | 1,089 (8.4%) | 354 (7.3%) | 735 (9.0%) | |
| Nigeria | | 477 (3.7%) | 105 (2.2%) | 372 (4.6%) | |
| Pakistan | | 1,664 (13%) | 642 (13%) | 1,022 (13%) | |
| Palestine | | 1,349 (10%) | 325 (6.7%) | 1,024 (13%) | |
| Saudi Arabia | | 245 (1.9%) | 86 (1.8%) | 159 (1.9%) | |
| Sudan | | 368 (2.8%) | 262 (5.4%) | 106 (1.3%) | |
| Syria | | 1,827 (14%) | 723 (15%) | 1,104 (14%) | |
| UAE | | 200 (1.5%) | 49 (1.0%) | 151 (1.8%) | |
| Yemen | | 1,692 (13%) | 605 (12%) | 1,087 (13%) | |
| Educational level | 13,030 | | | | <0.001** |
| Illiterate | | 145 (1.1%) | 36 (0.7%) | 109 (1.3%) | |
| Primary | | 183 (1.4%) | 35 (0.7%) | 148 (1.8%) | |
| Preparatory | | 379 (2.9%) | 70 (1.4%) | 309 (3.8%) | |
| Secondary | | 1,365 (10%) | 164 (3.4%) | 1,201 (15%) | |
| University | | 10,958 (84%) | 4,550 (94%) | 6,408 (78%) | |
| Marital status | 13,030 | | | | <0.001** |
| Married | | 4,196 (32%) | 1,336 (28%) | 2,860 (35%) | |
| Single | | 8,247 (63%) | 3,344 (69%) | 4,903 (60%) | |
| Widowed | | 246 (1.9%) | 70 (1.4%) | 176 (2.2%) | |
| Divorced | | 341 (2.6%) | 105 (2.2%) | 236 (2.9%) | |
| Occupational status | 13,030 | | | | <0.001** |
| Employed | | 3,946 (30%) | 1,951 (40%) | 1,995 (24%) | |
| Unemployed | | 6,968 (53%) | 2,268 (47%) | 4,700 (57%) | |
| Retired | | 344 (2.6%) | 72 (1.5%) | 272 (3.3%) | |
| Self-employed | | 1772 (14%) | 564 (12%) | 1208 (15%) | |

* significant p-value <0.05

** highly significant p-value <0.001

(p<0.001). Country-level subgroup analysis of screening barriers was not performed, as the primary aim of the study was to provide a combined multi-country overview. In addition, the participating countries share broadly similar cultural and lifestyle characteristics, which supports the presentation of aggregated findings.

As shown in Table 4, the multivariate logistic regression model revealed that individuals aged 25–49 years had higher odds of acquiring higher knowledge (aOR = 1.23, CI = 1.10-1.37, p<0.001). Females were less likely to have higher knowledge levels compared to males (aOR = 0.86, CI = 0.79-0.93, p<0.001). Regarding education, individuals with

**Table 2. Awareness Regarding Colorectal Cancer Symptoms.**

| Characteristic | Number | Overall Number, (%) | Working or studying in healthcare, (%) | Not working or studying in healthcare, (%) | p-value |
|---|---|---|---|---|---|
| Bleeding from the back passage | 13,030 | 8,526 (65%) | 3,738 (77%) | 4,788 (59%) | <0.001** |
| Persistent pain in the abdomen | 13,030 | 7,611 (58%) | 3,277 (67%) | 4,334 (53%) | <0.001** |
| Change in bowel habits over weeks | 13,030 | 7,614 (58%) | 3,499 (72%) | 4,115 (50%) | <0.001** |
| Feeling bowel doesn't empty completely | 13,030 | 5,691 (44%) | 2,758 (57%) | 2,933 (36%) | <0.001** |
| Blood in stool | 13,030 | 8,345 (64%) | 3,611 (74%) | 4,734 (58%) | <0.001** |
| Pain in the back passage | 13,030 | 5,083 (39%) | 2,367 (49%) | 2,716 (33%) | <0.001** |
| Lump in the abdomen | 13,030 | 9,134 (70%) | 3,774 (78%) | 5,360 (66%) | <0.001** |
| Tiredness/anemia | 13,030 | 6,634 (51%) | 3,297 (68%) | 3,337 (41%) | <0.001** |
| Unexplained weight loss | 13,030 | 8,754 (67%) | 3,967 (82%) | 4,787 (59%) | <0.001** |

** highly significant p-value <0.001

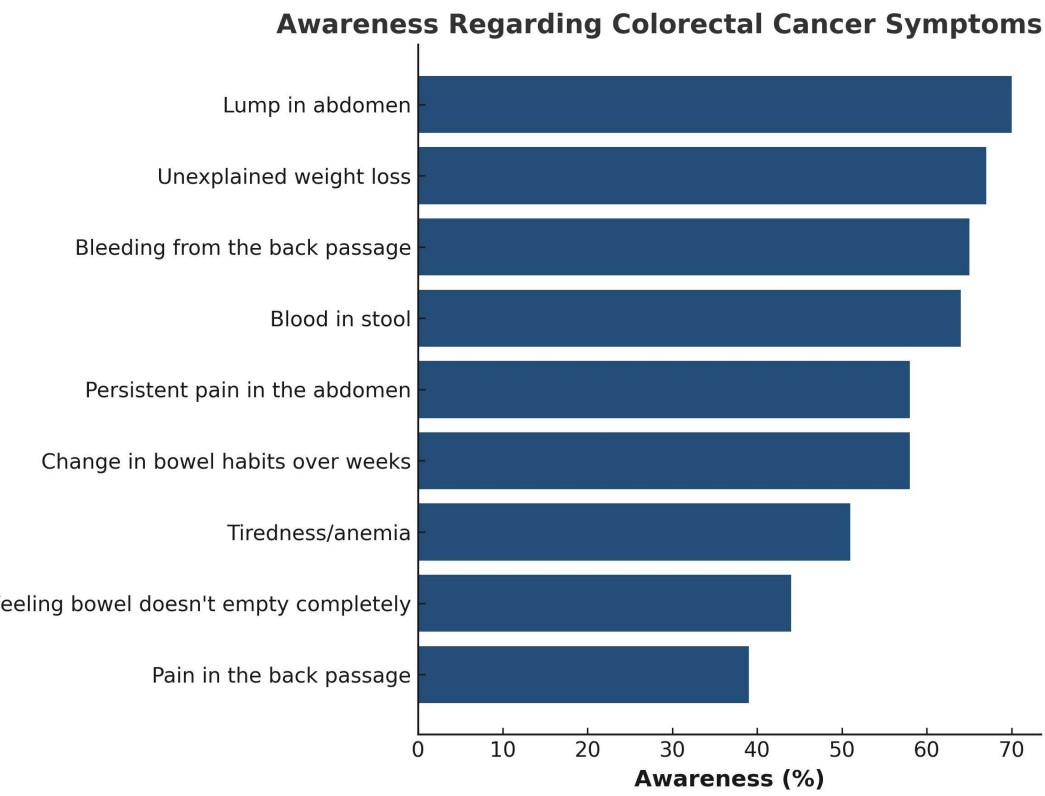

**Fig 1. The recognition percentages of CRC symptoms among participants.**

preparatory/secondary education and those with primary or no education had significantly lower odds of high knowledge compared to university graduates (aOR = 0.51, CI = 0.44-0.59, p < 0.001). As expected, those who are not working in the healthcare sector showed a significant decrease in odds compared to those who are in the healthcare workforce (aOR = 0.37, CI = 0.34-0.40, p < 0.001).

Model fit was assessed using the Hosmer–Lemeshow goodness-of-fit test and AIC. Multicollinearity was evaluated using variance inflation factors (VIF); no violations of model assumptions were identified.

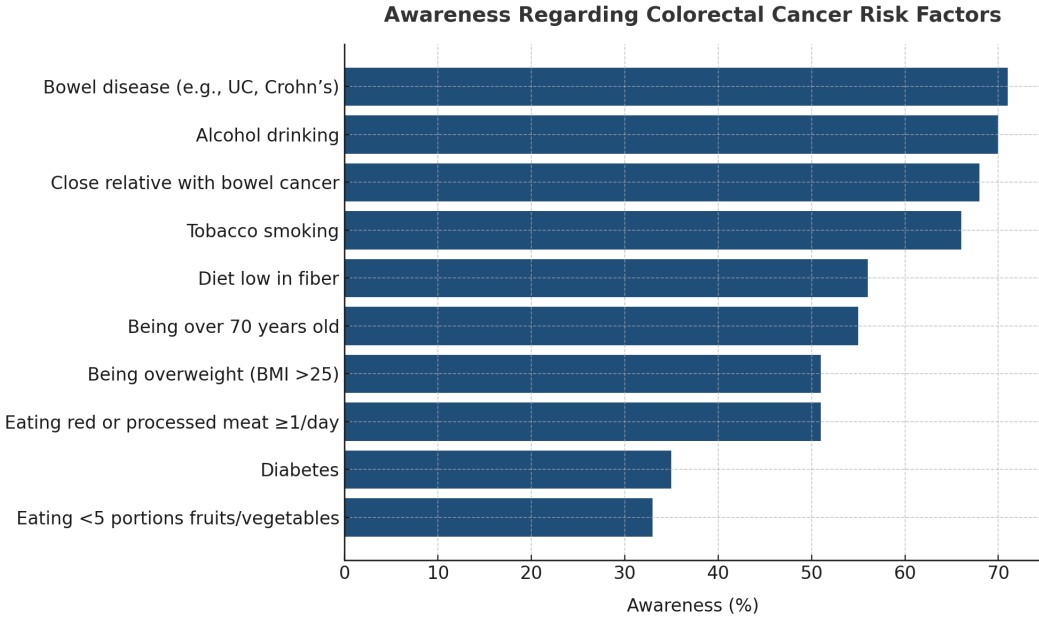

**Fig 2. The recognition percentages of CRC risk factors among participants.**

**Table 3. Percentage distribution of participants according to barriers.**

| Barrier | Strongly Agree (%) | Somewhat Agree (%) | Neutral (%) | Somewhat Disagree (%) | Strongly Dis-agree (%) | Do Not Know (%) | p-value |
|---|---|---|---|---|---|---|---|
| Not at risk due to the absence of symptoms | 33 | 30 | 14 | 13 | 5.6 | 4.8 | <0.001** |
| Not at risk due to a healthy lifestyle | 30 | 35 | 16 | 11 | 4.5 | 3.9 | <0.001** |
| Not at risk due to the absence of family history | 25 | 30 | 18 | 16 | 6.8 | 4.8 | <0.001** |
| Lack of time | 23 | 24 | 21 | 18 | 10 | 3.8 | <0.001** |
| Lack of reminders | 26 | 26 | 21 | 15 | 7.9 | 4.0 | <0.001** |
| Fear of diagnosis | 35 | 26 | 15 | 13 | 8.1 | 3.1 | <0.001** |
| Fear of tests | 34 | 26 | 15 | 14 | 7.9 | 3.4 | <0.001** |
| Embarrassment during the test | 32 | 26 | 16 | 15 | 8.1 | 3.6 | <0.001** |
| Inconvenience of the test | 28 | 26 | 20 | 14 | 7.0 | 5.0 | <0.001** |
| Doubt about the effectiveness of screening | 20 | 22 | 22 | 19 | 11 | 5.6 | <0.001** |
| The far distance of the screening center | 20 | 22 | 21 | 19 | 12 | 5.7 | <0.001** |

** highly significant p-value <0.001

## 4. Discussion

Among all the types of cancer, CRC is ranked as the second type that causes mortality globally [1]; however, this ranking may differ by country or region. By 2030, the incidence and mortality of CRC are expected to increase by 60% [24]. However, mortality rates of CRC can be reduced through frequent screening and prevention [25]. This highlights the importance of raising the level of CRC awareness, its symptoms, and risk factors. Our study focuses on assessing the level of knowledge about CRC among the population of nine different countries (Jordan, Egypt, Iraq, Bahrain, Syria, Pakistan, Palestine, Yemen, and Nigeria) and identifying the barriers to CRC screening among the participants.

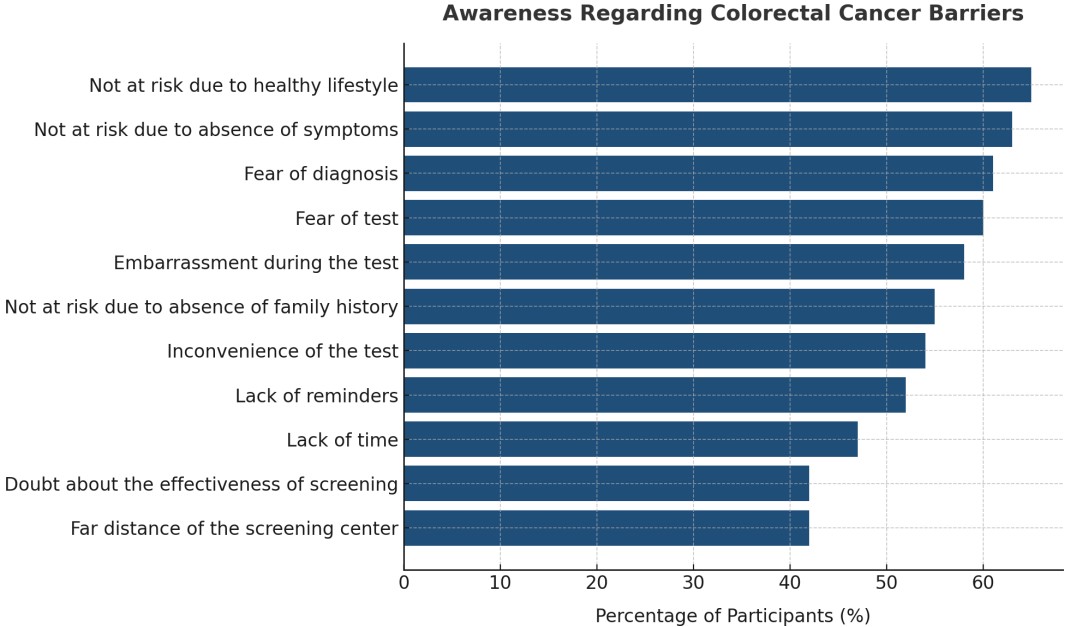

**Fig 3. The recognition percentages of CRC screening barriers among participants.**

**Table 4. Demographic Characteristics and Knowledge Levels: Univariate and Multivariate Odds Ratios logistic regression model.**

| Variable | Category | Low Knowledge n (%) | High Knowledge n (%) | Unadjusted OR (95% CI) | p-value | Adjusted OR (95% CI) | p-value |
|---|---|---|---|---|---|---|---|
| **Age group** | <25 (Ref) | 4874 (70.6) | 2030 (29.4) | Reference | — | Reference | — |
| | 25–49 | 3559 (66.6) | 1782 (33.4) | 1.20 (1.11–1.30) | <0.001** | 1.23 (1.10–1.37) | <0.001** |
| | ≥50 | 615 (78.3) | 170 (21.7) | 0.66 (0.55–0.79) | <0.001** | 1.07 (0.86–1.35) | 0.532 |
| **Gender** | Male (Ref) | 3725 (66.9) | 1842 (33.1) | Reference | — | Reference | — |
| | Female | 5323 (71.3) | 2140 (28.7) | 0.81 (0.75–0.88) | <0.001** | 0.86 (0.79–0.93) | <0.001** |
| **Education** | University (Ref) | 7275 (66.4) | 3683 (33.6) | Reference | — | Reference | — |
| | Prep/Secondary | 1481 (84.9) | 263 (15.1) | 0.35 (0.31–0.40) | <0.001** | 0.51 (0.44–0.59) | <0.001** |
| | Illiterate/Primary | 292 (89.0) | 36 (11.0) | 0.24 (0.17–0.34) | <0.001** | 0.30 (0.20–0.42) | <0.001** |
| **Marital status** | Married (Ref) | 3017 (71.9) | 1179 (28.1) | Reference | — | Reference | — |
| | Single | 5583 (67.7) | 2664 (32.3) | 1.22 (1.13–1.33) | <0.001** | 1.21 (1.09–1.35) | <0.001** |
| | Widowed/Divorced | 448 (76.3) | 139 (23.7) | 0.79 (0.65–0.97) | 0.025* | 1.08 (0.86–1.34) | 0.512 |
| **Occupation** | Employed (Ref) | 2371 (60.1) | 1575 (39.9) | Reference | — | Reference | — |
| | Unemployed | 4977 (71.4) | 1991 (28.6) | 0.60 (0.55–0.65) | <0.001** | 0.77 (0.69–0.85) | <0.001** |
| | Retired | 270 (78.5) | 74 (21.5) | 0.41 (0.31–0.53) | <0.001** | 0.69 (0.51–0.92) | 0.013* |
| | Self-employed | 1430 (80.7) | 342 (19.3) | 0.36 (0.31–0.41) | <0.001** | 0.45 (0.39–0.52) | <0.001** |
| **Healthcare worker** | Yes (Ref) | 2617 (53.9) | 2238 (46.1) | Reference | — | Reference | — |
| | No | 6431 (78.7) | 1744 (21.3) | 0.32 (0.29–0.34) | <0.001** | 0.37 (0.34–0.40) | <0.001** |

OR = Odds Ratio

CI = Confidence Interval

* significant p-value <0.05

** highly significant p-value <0.001

Among 13,030 participants from nine countries, most of the respondents were females rather than males (57% vs. 43%), and this is aligned with many similar cross-sectional studies conducted for CRC [22,23,26,27]. Additionally, most of the participants of our study were from Syria (1,827), followed by Yemen (1.692), Pakistan (1,664), and Egypt (1,513). Notably, the number of respondents from Syria who participated in our study is more than those included in Almoshanatef et al. and Zayegh et al.'s cross-sectional studies that assessed the knowledge level of CRC among the Syrian population (702 and 824, respectively) [26,27].

This study revealed that among the nine participating countries, the knowledge level about CRC is probably low, with only 31% of the participants having a high level of knowledge. This result is compatible with most of the findings in the literature. In Qatar, a study reported that the awareness level of CRC symptoms and risk factors among 448 participants was low, with an overall awareness score of 9.03/20 (SD ± 5.5) [22]. Moreover, a national cross-sectional study assessing the correlation between CRC level of knowledge and attitudes towards CRC screening in Saudi Arabia among 5720 participants reported a CRC awareness score of 11.05 (SD 4.4, range 1–23) [28]. Besides that, 4623 participants in another national cross-sectional study conducted in Palestine had only a 40% awareness level for CRC, with "lump in the abdomen" as the most frequently identified CRC symptom among the participants (74%) [29]. Additionally, a study in Lebanon found that only 17.2% and 31.5% were aware of CRC risk factors and symptoms, respectively [30]. In Egypt, a recent study found that only 29% of participants recognized key CRC risk factors [31]. Also, in Iran, the mean awareness score of CRC risk factors was 3.63/10, indicating a significant knowledge gap in the Iranian population [32]. Also, a study conducted in Pakistan showed that the mean knowledge of participants regarding CRC risk factors and symptoms was 7.52 out of 14 [33].

In Kuwait, awareness levels were slightly better, with 54.8% recognizing blood in stools and 59.4% identifying unexplained weight loss as CRC symptoms [34].

Among all the suggested CRC symptoms in our questionnaire, a lump in the abdomen or tummy was the most recognized CRC symptom (70%), followed by unexplained weight loss (67%) and bloody stool (64%) among the participants. These findings are precisely aligned with the findings of some of the previous studies but with variable CRC awareness levels [22,29,34].

As reported in our study, the presence of bowel diseases such as ulcerative colitis and Crohn's disease was the most recognizable risk factor for CRC among the respondents, followed by drinking alcohol. This is different from the survey of Qatar, which reported that the "daily eating of processed meat" is the risk factor with the highest awareness level among the represented participants [22]. On the other hand, a survey conducted in Palestine reported that the most recognized non-modifiable CRC risk factor was having a bowel disease (IBD for example) [29]. Also, a study in Lebanon revealed "bowel disease" as the second most identified CRC risk factor among the participants, while "red meat" was again the most recognizable CRC risk factor [30].

Although many previous studies share a similar aim to ours, differences in study populations limit direct comparability. For example, Al-Dahshan et al. and El Muhtaseb MS et al. focused on a specific age range (50–74 years) [22,35] while Zayegh et al. and Aga SS et al. recruited only healthcare students from Aleppo University, Syria, and King Saud Bin Abdulaziz University for Health Sciences, Saudi Arabia [27,36].In contrast, our study included a multi-country sample of general adults, which may explain differences in observed knowledge levels and screening behaviors.

Frequent CRC screening is crucial for early detection and treatment of the disease; however, according to many studies, there is generally a negative attitude towards CRC screening, especially among MENA region countries [37]. Our study assesses the barriers that hinder the participants from frequent CRC screening among the nine countries included. The most frequent barrier among the participants is the misconception that there is no risk of disease development if there are no symptoms and/or a family history, and if there is a healthy lifestyle. Test-related barriers, including test fear, embarrassment, and inconvenience, were also common. In comparison to the reported findings in the literature, Al-Dahshan et al. identified the same barriers among 188 Arab participants in addition to the barrier of "doubt about the effectiveness of

CRC screening methods. [23]". Another study conducted in Saudi Arabia reported that the absence of symptoms, fear of procedure, and results were the most common barriers among the Saudi population [38]. Besides, in a study conducted in Lebanon, the most common barrier against CRC screening was the lack of knowledge about the importance of CRC screening, in addition to the misconception that the absence of family history minimizes the risk of disease development [30]. All the reported barriers against CRC screening highlight the importance of raising awareness of CRC to clear the idea of CRC fatality and hopelessness, in addition to emphasizing that early detection of CRC can alleviate CRC-accompanied morbidity and mortality. Concerning this, a study revealed the importance of presenting an educational curriculum explaining CRC among university students and how its effect lasted for six months [39].

In Iraq, awareness and screening uptake remain low, largely due to limited access to screening programs and financial barriers, with many individuals lacking knowledge about CRC risk factors and participating infrequently in screening [40]. Similarly, studies from Turkey and the UAE have identified the failure of healthcare providers to actively recommend or offer screening as a major barrier, contributing to low participation rates [41,42], In contrast, countries such as Norway and the UK have successfully increased CRC screening uptake through the implementation of national, state-funded screening programs [43,44].These examples highlight the potential benefits of establishing national screening initiatives in other contexts. Additionally, introducing the fecal immunochemical test (FIT) as a non-invasive screening alternative could encourage participation among individuals reluctant to undergo colonoscopy.

Unlike some previous studies [29,45] our study found that males were more likely to have a good awareness level about colorectal cancer than females. This may reflect cultural factors influencing health literacy and information-seeking behaviors differently by gender. Additionally, the online recruitment strategy may have introduced selection bias, favoring males who are more active on social media or more likely to complete online surveys. Although we did not conduct country-specific analyses of gender differences, the pooled sample included countries with broadly similar cultural practices and healthcare contexts. These considerations should be taken into account when interpreting the gender-related findings, and future research could explore how gender differences vary across individual countries.

In our study, participants aged 25–49 were most likely to have a good awareness level about colorectal cancer. Comparisons with previous literature are limited by differences in the age ranges studied; for example, a study in Bahrain reported higher CRC awareness among individuals older than 49 years [45]. Similarly, a study conducted in Iran demonstrated a higher prevalence of colorectal cancer among individuals aged over 40 years [46]. Consistent with expectations, higher educational attainment was associated with greater CRC knowledge. Although results vary across studies, research from Qatar and Bahrain reported similar associations between university-level education and CRC awareness [23,45]. Healthcare professionals in our sample demonstrated higher CRC knowledge compared with non-healthcare participants. Interestingly, a study in Jordan assessing pharmacists' awareness of CRC found a surprisingly low level of knowledge regarding early detection and appropriate screening methods [47].

One of the strengths of this study is the use of a pre-existing validated questionnaire available in both English and Arabic to ensure the feasibility of understanding the questions. Moreover, a huge sample size from nine different countries participated in the study, giving a good presentation about general CRC awareness among the public. However, the inclusion of nine countries in one analysis can be one of the limitations because there is a cultural difference between the included countries that can affect the level of participants' attitudes and awareness towards CRC screening. Another strength is our high reliability and accuracy in data collection and analysis. That is why we believe that the current study's findings are authentic and comparable to the existing studies in the literature.

Despite providing valuable insights into public awareness of colorectal cancer (CRC) symptoms, risk factors, and screening barriers, this study has several limitations. First, the online, self-administered survey required internet access, digital literacy, and proficiency in English or Arabic, which may have underrepresented older adults, rural populations, and individuals with limited education or internet access, introducing selection bias and limiting generalizability. Second, the use of convenience sampling may have led to overrepresentation of more health-aware individuals, and non-response

bias cannot be excluded as characteristics of non-respondents were not assessed. Third, although all adults aged ≥18 years were eligible, CRC screening is primarily relevant to older populations. The inclusion of younger adults may limit the direct applicability of screening-related findings to screening-eligible groups. Age categorization was based on earlier guidelines recommending screening from age 50, whereas current guidelines recommend starting at age 45 [13]. Subgroup analyses restricted to screening-age individuals were not performed because the study was not powered for this purpose; however, younger adults were intentionally included to assess general CRC awareness and inform early educational efforts. Fourth, data were self-reported and may be subject to recall or social desirability bias, potentially overestimating knowledge or underreporting barriers. Finally, the absence of country-stratified analyses limits the ability to draw conclusions at the individual country level. Additionally, although most participating countries are Muslim-majority, religion-specific dietary practices or cultural behaviors were not assessed; therefore, associations between colorectal cancer knowledge or risk and religious or cultural practices could not be evaluated. Future research incorporating country-specific analyses and healthcare system characteristics would help clarify regional variations in knowledge and screening behaviors.

## 5. Conclusion

In conclusion, this study's findings demonstrate that public awareness and knowledge about colorectal cancer are low in the surveyed population. This shows an urgent need for educational programs to raise awareness of colorectal cancer symptoms and risk factors. Reducing the barriers that keep people away from getting medical attention should also be a priority of these initiatives, as this may result in earlier detection and better disease outcomes.

## Supporting information

**S1 Table. Awareness regarding colorectal cancer risk factors.**
(DOCX)

**S1 Text. EARG Collaborators.**
(DOCX)

## Acknowledgments

We want to acknowledge Ahmad Alkhader, Miriam Ashi, Adnan Eid, Yasmin Ahmadi, Abrar Ghaith, Emad Alkadri, Shurouq Seder, Diaa Dawod, Bello Saifullah Muhammad, Barakat Olajumoke Kolawole, Ameen Alwossabi, Rachad Alnamer, and Israa Amad Outob for their effort in collecting data.

## Author contributions

**Conceptualization:** Aseel Ghanayem, Husam Abu Dawood, Hanan Khaled Almokdad, Nael Kamel Eltewacy.

**Data curation:** Aseel Ghanayem, Mandy Elewa, Tarek A. Owais, Rawan Ghanayem, Hanan Khaled Almokdad.

**Formal analysis:** Husam Abu Dawood, Mandy Elewa.

**Funding acquisition:** Aseel Ghanayem, Husam Abu Dawood, Mandy Elewa, Tarek A. Owais.

**Investigation:** Aseel Ghanayem, Husam Abu Dawood, Mandy Elewa, Tarek A. Owais, Rawan Ghanayem, Hanan Khaled Almokdad.

**Methodology:** Aseel Ghanayem, Husam Abu Dawood, Mandy Elewa, Tarek A. Owais, Rawan Ghanayem, Hanan Khaled Almokdad.

**Project administration:** Aseel Ghanayem, Tarek A. Owais, Rawan Ghanayem.

**Resources:** Mandy Elewa, Tarek A. Owais, Hanan Khaled Almokdad.

**Software:** Mandy Elewa, Tarek A. Owais, Hanan Khaled Almokdad.

**Supervision:** Aseel Ghanayem, Nael Kamel Eltewacy.

**Validation:** Aseel Ghanayem, Rawan Ghanayem, Nael Kamel Eltewacy.

**Visualization:** Aseel Ghanayem, Husam Abu Dawood, Rawan Ghanayem, Nael Kamel Eltewacy.

**Writing – original draft:** Aseel Ghanayem, Husam Abu Dawood, Tarek A. Owais, Rawan Ghanayem, Hanan Khaled Almokdad, Nael Kamel Eltewacy.

**Writing – review & editing:** Husam Abu Dawood, Nael Kamel Eltewacy.

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
