## [Decision Letter · Decision Letter 0]

26 Dec 2025

PGPH-D-25-03160

Public Awareness of Colorectal Cancer Symptoms and Risk Factors, and Exploring Screening Barriers Across Nine Countries: A Multi-Center Cross-Sectional Study

Dear Dr. Husam Abu Dawood

Thank you for submitting your manuscript to PLOS Global Public Health. After careful consideration, we feel that it has merit but does not fully meet PLOS Global Public Health’s publication criteria as it currently stands. Therefore, we invite you to submit a revised version of the manuscript that addresses the points raised during the review process.

The paper presents important findings across countries on an important public health topic. While these countries are predominantly Muslim majority, the authors do not comment on whether dietary or cultural factors sees a higher CRC risk in this population. If this not a consideration, the authors can focus on the reviewer reports and addressing my comment belowPlease see the reviewer reports and address each of them. These will improve the overall readability of the paper. I also suggest that the authors add table 3 as a supplementary table and table 4 should be subsumed into the text and removed. Table 5 mainly presents very similar percentages across rows and can be better summarized in text as well. Table 6 has formatting and presentation issues that the reviewers have also raised. Please revise table 6 in line with standard presentation of unadjusted and adjusted results.

We look forward to receiving your revised manuscript.

Kind regards,

Danish Ahmad, MBBS,MSc,MNAMS,PhD,IP-FPH(UK),FRCP(Edin),FRCP(Lon)

Academic Editor

Journal Requirements:

1.  In the online submission form, you indicated that All data generated or analyzed are included in this article. Original data set/ raw data are available from the corresponding author on reasonable request.

3. Uploaded as supplementary information.

2. We have noticed that you have uploaded Supporting Information files, but you have not included a list of legends. Please add a full list of legends for your Supporting Information files after the references list.

Reviewers' comments:

Reviewer's Responses to Questions

**Comments to the Author**

1. Does this manuscript meet PLOS Global Public Health’s publication criteria?

Reviewer #1: Yes

Reviewer #2: Yes

Reviewer #3: Yes

Reviewer #4: Yes

2. Has the statistical analysis been performed appropriately and rigorously?

Reviewer #1: Yes

Reviewer #2: Yes

Reviewer #3: Yes

Reviewer #4: Yes

3. Have the authors made all data underlying the findings in their manuscript fully available (please refer to the Data Availability Statement at the start of the manuscript PDF file)?

Reviewer #1: Yes

Reviewer #2: Yes

Reviewer #3: No

Reviewer #4: Yes

4. Is the manuscript presented in an intelligible fashion and written in standard English?

Reviewer #1: Yes

Reviewer #2: Yes

Reviewer #3: Yes

Reviewer #4: No

Reviewer #1: The topic is very interesting. The sample size is very large. The survey covers several countries.

Introduction

In the introduction, it is important to mention similar work that has been done in this area. It is also necessary to focus on the knowledge gap to be filled regarding the phenomenon being studied.

Method

Why the age limit of 18? This needs to be explained.

You explained how the sample size was obtained. However, it is necessary to explain how the individuals were selected. Since the questionnaire is self-administered, there is a possibility of selection bias, such as the exclusion of people who are illiterate.

Results

Online questionnaires are often associated with non-response. Are there any non-respondents, and what is the profile of these non-respondents?

In the results, it is also important to look for barriers specific to each country. If there are differences between countries.

Discussion

We need to discuss selection biases that may be related to non-respondents, education level, internet access, and sampling.

Reviewer #2: Topic: “Public Awareness of Colorectal Cancer Symptoms and Risk Factors, and Exploring Screening Barriers Across Nine Countries: A Multi-Center Cross-Sectional Study.”

--Manuscript Draft Reviewed—19 December 2025

Manuscript Number: PGPH-D-25-03160

Correction to the manuscript:

Please note: The Manuscript Body Formatting Guidelines were not followed. Paragraphs should also be indented. Numeric numbers (for example, 1) should start at Abstract.

Methods: I would rewrite the methods to read: “A cross-sectional study was conducted across multiple centers in nine countries from February 1, 2024, through April 3, 2024, targeting individuals aged 18 years and older. Data were collected via an online questionnaire in collaboration with national leaders and local team members in each country. To examine factors influencing participants’ knowledge, awareness, and screening barriers, logistic regression analyses were employed.”

Results: I would rewrite the methods to read: “Among the 13,030 participants, awareness of symptoms was highest for "lump in the abdomen" (70%) and lowest for “back pain” (39%). The most recognized risk factor was “history of bowel disease” (71%), and the least recognized was “low fruit/vegetable intake” (33%). The primary barrier to screening was the belief in low personal risk due to the absence of symptoms (63%). Multivariate logistic regression identified working in the healthcare sector as the strongest predictor of high knowledge (aOR = 2.70, 95% CI: 2.50-2.94). Higher knowledge was also associated with males, older age (25-49 years), and university-level education (all p < 0.001).

Line 62 & 63: “Around the world, colorectal cancer (CRC) is the third most common type of cancer and the second leading cause of cancer-related deaths, according to the World Health Organization (WHO, year?).” It is primarily … “Are you talking about CRC?” Please correct to read "CRC."

Line 64: In 2020 alone. This still means in 2020. Please remove the word “alone.”

Line 72: “ … face a growing burden.” (Citation needed here).

Line 74: Should be left aligned.

Lines 90-91: “ … countries still lack well-established national screening programs.” Who do you know? Please cite.

Line 92-93: What are those current guidelines? Please cite?

Line 94: Should be left aligned.

Line 97: Several barriers hinder individuals from undergoing regular screening for CRC, such as ….. ? Please state these barriers.

Line 101: The study aim should be aligned with the current topic. For example, “Public Awareness of Colorectal Cancer Symptoms and Risk Factors, and Exploring Screening Barriers Across Nine Countries (Jordan, Egypt, Iraq, Bahrain, Syria, Pakistan, Palestine, Yemen, and Nigeria).” This is called research alignment. Please rectify.

Line 111: Is it that you are doing “multi-national” or “multi-centered”? Please be specific.

Eligibility criteria:

Lines 117-119: The eligibility criteria should be specific as to who is included and who is excluded from the study. This area is not specific. This area is indecisive.

Samling:

Lines 123-125: 385 * 9 = 3,465 participants eligible for the study. I assume that these are eligible to participate in the study across nine countries. How many were ineligible? What was the population size? How many had an incomplete survey (started but did not submit)?

Line 129-130: What was the validated questionnaire? Please state.

Line 131: You “also” created a self-administered survey.

Line: 172-175: You state nine countries but only had approval from Egypt, Yemen, Iraq, and Syria. What happened to the approval from Jordan, Bahrain, Pakistan, Palestine, and Nigeria?

Line: 180-182: Categorical data is treated as the dependent variable in most analyses, but not in all cases. Demographic variables are also treated as independent variables; however, the verbiage stated is dependent on the analyses undertaken (logistic, multinomial regression, etc.).

Lines 200-202: A total of 16,940 participants were invited to participate in the study. However, 3,910 responses were excluded due to ineligible demographics (age restrictions, residency outside targeted countries, or duplicate entries), resulting in a final sample size of 13,030 participants. Some clarification is needed here. Please refer to lines 123-125.

Line 207: I see below in the table, but you mentioned males here 43% (n = 5,567). How many were females (in the same line as the males)? What percentage (57%, n = 7,463)?

Line 220: Caps for Table 2.

Line 30: Corrections to Table 3 remove the brackets for ease of reading.

Line 247: I would recommend that tables stay on separate pages—not hanging.

Line 253: (Table 5). Remove brackets.

Line 292: “Among all the types of cancer, CRC is ranked as the second type that causes mortality.” According to whom? Please state.

Line 292-324: Areas should be consistent with spaces.

Reviewer #3: Data Availability: The current data availability statement does not meet PLOS standards. Authors should deposit the anonymized dataset in a public repository (e.g., Figshare, Dryad, or OSF) to ensure full transparency and reproducibility.

Country Selection and Justification: The rationale for selecting these specific nine countries is unclear. The abstract mentions nine countries, but Table 1 shows 13 countries (including Bangladesh, Saudi Arabia, Sudan, and UAE not mentioned in the methods). This discrepancy needs clarification. Please explain:

Why these countries were selected

The inclusion criteria for countries

How the additional four countries were incorporated

Sample Size Calculation: While you mention needing 385 responses per country using Raosoft calculator, several countries fall below this threshold (e.g., UAE=200, Sudan=368, Bangladesh=256). Please address:

How this affects the representativeness of findings

Whether country-specific analyses are adequately powered

Consider stratified analysis or acknowledging this as a limitation

Knowledge Scoring Methodology: The 70% cut-off for "high knowledge" appears arbitrary. Please provide:

Justification for this specific threshold

Whether this has been validated in previous studies

Sensitivity analysis with different cut-offs

Ethical Approval Inconsistency: Ethics approval was obtained from only 4 countries (Egypt, Yemen, Iraq, Syria) out of 9-13 countries. Please clarify:

Why ethics approval wasn't obtained from all participating countries

Whether institutional review was conducted for countries not listed

How ethical standards were maintained across all sites

Statistical Methods:

The chi-square tests comparing healthcare vs. non-healthcare workers are appropriate, but consider adding effect sizes (e.g., Cramér's V) to complement p-values

For the logistic regression, consider reporting model fit statistics (e.g., AIC, Hosmer-Lemeshow test)

Consider testing for multicollinearity among predictors in the multivariate model

Discussion of Country-Specific Variations: Given the cultural and healthcare system differences across countries, the lack of country-stratified analysis is a limitation. Consider:

Adding supplementary tables showing country-specific knowledge levels

Discussing how healthcare infrastructure differences might explain variations

Acknowledging regional heterogeneity more explicitly

Gender Findings: The finding that males have higher knowledge contradicts some cited studies. The discussion mentions this briefly (line 373-374) but doesn't explore potential explanations. Consider discussing:

Cultural factors affecting health literacy by gender

Selection bias in your sample (online recruitment may favor certain demographics)

How this varies by country

Age Categorization: The age grouping (<25, 25-49, ≥50) seems inconsistent with CRC screening guidelines (typically 45/50-75 years). Consider:

Regrouping to align with screening-eligible populations

Separate analysis for screening-eligible age groups

Justification for current categories

Abstract: Line 46-47 states working in healthcare is the "strongest predictor" (aOR=2.70) but doesn't compare this to the reference category clearly. Specify that this is compared to non-healthcare workers.

Introduction:

Line 67: "More than 80%" - please provide the specific reference

Line 92-94: Update screening age recommendations as recent guidelines now suggest starting at 45 years for average-risk individuals

Methods:

Line 123-125: "Convenience sampling" - acknowledge this as a limitation affecting generalizability

Line 131-132: Specify which modifications (if any) were made to the validated questionnaire

Line 165-166: Clarify how duplicate entries were identified

Results:

Table 1: The p-value for age (0.027) is significant at p<0.05 but text states "only age showed no significant association" - this appears contradictory

Line 243-244: "31% demonstrated high knowledge" - provide breakdown by country as this likely varies substantially

Discussion:

Line 292: "CRC is ranked as the second type that causes mortality" - globally, yes, but this may differ by country/region

Line 337-342: The comparison to studies with different populations is appropriate, but acknowledge how this limits direct comparability

Lines 362-371: The addition about Iraq, Turkey, UAE, and Norway is valuable but feels somewhat disconnected. Consider integrating more smoothly or expanding

Line 373-386: Good comparison with literature, but consider organizing by theme rather than country

Limitations Section (Lines 396-402):

Excellent acknowledgment of online survey bias

Add: convenience sampling, country selection, lack of country-specific analysis

Add: potential recall bias in self-reported knowledge

Add: cross-sectional design preventing causal inference

Language and Presentation:

Line 68: "changes in lifestyle and dietary habits" - be more specific

Line 337: "It is worth mentioning" - remove informal phrase

Consider consistent terminology: "bowel cancer" vs "colorectal cancer"

Figures:

Figures 1-3 are clear and well-presented

Consider adding error bars or confidence intervals

In Figure 3, consider reordering from highest to lowest percentage for easier interpretation

References:

Reference 2 appears incorrect - cites an antiemetics guideline rather than CRC epidemiology

Ensure all references are current (several from 2011-2017 may have updates)

Reviewer #4: I applaud the authors for such a great and important work.

1. The first three sentences of the introduction need references.

2. One of the limitations listed is having younger people respond to questions on limitations for screening thus misrepresenting the true picture. I think this limitation could be overcome if the authors choose to do a separate analysis of thus objective using respondents only within the age of screening.

**Do you want your identity to be public for this peer review?** For information about this choice, including consent withdrawal, please see our Privacy Policy

Reviewer #1: No

Reviewer #2: No

Reviewer #3: No

Reviewer #4: No

---

## [Decision Letter · Decision Letter 1]

29 Jan 2026

Public Awareness of Colorectal Cancer Symptoms and Risk Factors, and Exploring Screening Barriers Across Nine Countries: A Multi-National Cross-Sectional Study

PGPH-D-25-03160R1

Dear Dr. Abu Dawood,

We are pleased to inform you that your manuscript 'Public Awareness of Colorectal Cancer Symptoms and Risk Factors, and Exploring Screening Barriers Across Nine Countries: A Multi-National Cross-Sectional Study' has been provisionally accepted for publication in PLOS Global Public Health.

Best regards,

Julia Robinson

Executive Editor

Reviewer Comments (if any, and for reference):

Reviewer's Responses to Questions

**Comments to the Author**

Reviewer #2: All comments have been addressed

Reviewer #3: All comments have been addressed

Reviewer #4: All comments have been addressed

publication criteria?

Reviewer #2: Yes

Reviewer #3: Yes

Reviewer #4: Yes

3. Has the statistical analysis been performed appropriately and rigorously?

Reviewer #2: Yes

Reviewer #3: Yes

Reviewer #4: Yes

4. Have the authors made all data underlying the findings in their manuscript fully available (please refer to the Data Availability Statement at the start of the manuscript PDF file)?

Reviewer #2: Yes

Reviewer #3: Yes

Reviewer #4: Yes

5. Is the manuscript presented in an intelligible fashion and written in standard English?

Reviewer #2: Yes

Reviewer #3: Yes

Reviewer #4: Yes

Reviewer #2: The authors have comprehensively addressed all comments and concerns raised, and the manuscript titled “Public Awareness of Colorectal Cancer Symptoms and Risk Factors, and Exploring Screening Barriers Across Nine Countries: A Multi-National Cross-Sectional Study” has been substantially improved as a result.

Reviewer #3: no corrections

Reviewer #4: I have no further comments on the work.

**Do you want your identity to be public for this peer review?** For information about this choice, including consent withdrawal, please see our Privacy Policy

Reviewer #2: No

Reviewer #3: No

Reviewer #4: **Yes:** Faraja M. Magwesela
